# Anti-Citrullinated Protein Antibody Titers Are Independently Modulated by Both Disease Activity and Conventional or Biologic Anti-Rheumatic Drugs

**DOI:** 10.3390/diagnostics12071773

**Published:** 2022-07-21

**Authors:** Miren Uriarte Ecenarro, Daniel Useros, Aranzazu Alfranca, Reyes Tejedor, Isidoro González-Alvaro, Rosario García-Vicuña

**Affiliations:** 1Rheumatology Service, Hospital Universitario La Princesa, IIS-Princesa, 28006 Madrid, Spain; miren_uriarte@hotmail.com; 2Internal Medicine Service, Hospital Universitario La Princesa, IIS-Princesa, 28006 Madrid, Spain; dani.mithrandir@gmail.com; 3Immunology Service, Hospital Universitario La Princesa, IIS-Princesa, 28006 Madrid, Spain; mariaaranzazu.alfranca@salud.madrid.org (A.A.); reyes.tejedor@salud.madrid.org (R.T.); 4Department of Medicine, Faculty of Medicine, Autonomous University of Madrid, 28029 Madrid, Spain

**Keywords:** early arthritis, Rheumatoid arthritis, ACPA, biomarkers, disease activity, disease-modifying anti-rheumatic drugs (DMARDs), therapies, clinical and analytical tools

## Abstract

This study aimed to analyze the factors that influence anti-citrullinated protein antibody (ACPA) titers in a seropositive early arthritis (EA) population under non-protocolized treatment with disease-modifying anti-rheumatic drugs (DMARDs). A total of 130 ACPA-positive patients from the PEARL (Princesa Early Arthritis Longitudinal) study were studied along a 5-year follow-up. Sociodemographic, clinical, and therapeutic variables, along with serum samples, were collected at five visits by protocol. Anti-cyclic citrullinated peptide 2 (CCP2) ACPA titers were measured by ELISA. The effect of different variables on anti-CCP2 titers was estimated using longitudinal multivariate analysis models, nested by visit and patient. Data from 471 visits in 130 patients were analyzed. A significant decrease in anti-CCP2 titers was observed at all time-points, compared to baseline, following the decline of disease activity. In the multivariate analysis, active or ever smoking was significantly associated with the highest anti-CCP2 titers while reduction in disease activity was associated with titer decline. After adjusting for these variables, both conventional synthetic (cs) and biologic (b) DMARDs accounted for the decline in anti-CCP2 titers as independent factors. Conclusion: In patients with EA, an early and sustained reduction in ACPA titers can be detected associated with the decline in disease activity, irrespective of the treatment used.

## 1. Introduction

Rheumatoid arthritis (RA) is an autoimmune systemic disease characterized by chronic inflammatory joint involvement, as well as extra-articular manifestations [1,2,3]. Contemporary treatment of early RA emphasizes the early introduction of DMARDs considering the “window of opportunity” timeframe [4], and the adjustment of therapy through the treat to target (T2T) and tight control strategies in order to reach remission, or, at least, low disease activity [5,6]. Despite the focus on early arthritis (EA) relies on abrogating inflammation [7], we still need diagnostic and prognostic biomarkers to tailor treatment options to maintain the balance between early control of the disease and avoiding overtreatment in patients with mild disease [8]. 

Autoantibodies such as rheumatoid factor (RF) and ACPA are detected in 70–80% of patients with established RA, but ACPA have shown to be more specific than RF in early RA, with comparable sensitivity [9]. Anti-CCP assays are the most widely used methods to study ACPA [9]. Several studies have demonstrated anti-CCP as reliable diagnostic and prognostic biomarkers in RA [10,11,12]. Furthermore, ACPA and their titers are the strongest predictors of joint structural progression [12,13,14,15] and are associated with a higher risk of developing interstitial lung disease [16] as well as increased cardiovascular morbidity and mortality [17,18,19]. 

Despite the robust evidence on the diagnostic and prognostic value of ACPA in RA, the association of ACPA titers with disease activity is less clear and has led to the concept of the “ACPA-paradox” [20]. The pathogenic mechanisms that link structural damage with autoantibodies through enhancing disease activity have been extensively reviewed and are well established in the case of RF, but not yet completely understood in the case of ACPA [20], which can induce bone damage even in absence of inflammation [21]. 

Epidemiological data have shown fluctuations in RF and ACPA titers along the disease course and decreases have been commonly linked to an improvement in disease activity [22,23]. However, the decrease in autoantibody titers has been reported to occur faster and to a greater extent for RF compared with ACPA [22,24,25]. Additionally, some evidence suggests that a shorter disease duration can independently affect the ACPA response under different csDMARDs [26] 

The effect of different treatments on ACPA levels is heterogeneous and has been mainly explored for bDMADRs, with reported benefits on clinical activity regardless of the decline in titers or seroconversion [23,27]. 

Concerning patients with EA, studies exploring the modulation of ACPA levels under csDMARDs have shown contradictory results [28,29,30,31,32,33] and the effect of individual drugs has been rarely assessed [26,34]. Regarding anti-CCP, most studies demonstrated that negative seroconversion is rare, the effect on titers is heterogeneous and their fluctuations do not consistently associate with disease activity nor seem to predict clinical or radiological outcomes [30,33,35]. Studies involving bDMARDs point toward a differential effect of abatacept (ABA) on ACPA levels, with a more consistent association between decline in ACPA titers and clinical response, compared with TNF-alpha antagonists [33,36].

Given contradictions or evidence gaps regarding modulation of ACPA levels, the present study aims to analyze which variables can influence the levels of anti-CCP ACPA in patients with EA, taking advantage of our PEARL (Princesa Early Arthritis Longitudinal) study, in which most patients have not received DMARD treatment at recruitment. 

## 2. Materials and Methods

### 2.1. Patient Population and Study Design

PEARL (Princesa Early Arthritis Register Longitudinal) study is a prospective longitudinal registry that includes all incident cases of patients with one or more swollen joints for less than a year referred to our EA Clinic. 

For this study, we analyzed only anti-CCP2+ patients, with at least 2 years of follow-up and with a diagnosis of RA according to ACR/EULAR 2010 criteria [37] or undifferentiated arthritis [38]. Patients diagnosed with other disorders during follow-up (septic arthritis, microcrystalline, osteoarthritis, spondyloarthritis, other autoimmune diseases or vasculitis) were excluded. We also excluded patients without serum samples collected at baseline or those without at least one follow-up visit. After written informed consent, samples and data from patients included in this study were provided by the Biobank Biobanco Hospital Universitario de La Princesa (ISCIII B.0000763) and they were processed following standard operating procedures with the appropriate approval of the Ethics and Scientific Committees.

The register started in 2000 and is still ongoing, it includes 5 structured visits (baseline, 6, 12, 24 and 60 months) in which socio-demographic, clinical, laboratory, therapeutic and radiological data as well as biological samples are systematically collected following a designed protocol. 

It is important to point out that there is no pre-established therapeutic protocol in PEARL study, so the decision on when and how to treat the patients during the follow-up relies on the responsible physicians from the rheumatology department (see Section 3.1 for description of treatments along the follow-up). Nevertheless, the register-specific evaluation visits are performed by only two rheumatologists (AMO, IG-A) in an attempt to achieve more accurate clinical evaluation, especially regarding joint counts. A more detailed description of the PEARL study has been previously published [39]. However, for this analysis we only used the visits from patients with anti-CCP2 positivity included in the database until April 2015. 

### 2.2. Variables

PEARL protocol includes: socio-demographic data; duration of illness up to recruitment; count of 28 tender and swollen joints, global assessment of the disease according to a visual analogue scale (VAS) by the patient (VGEP) and by the doctor (VGEM); physical function through Heath assessment questionnaire (HAQ); laboratory determinations, including acute phase reactants (ESR and CRP), RF; treatment variables, including dose and time of use of each DMARD and glucocorticoids.

Disease activity and therapeutic response were assessed using the composite indices DAS28 ESR [40] and Hospital Universitario la Princesa index (HUPI) [41].

Patients were categorized into responders or non-responders (YES/NO) according to moderate and good EULAR response criteria [42] and HUPI response criteria [43]. Remission was defined as DAS28 ESR < 2.6 [44] or HUPI <= 2 [43].

### 2.3. Anti-Citrullinated Protein Antibody Measurements

Baseline status (positive/negative) and anti-CCP2 concentration at each visit (6, 12, 24, and 60 months) were determined using an anti-CCP2 IgG ELISA (Euro Diagnostica Immunoscan CCPlus^®^, Arnhem, The Netherlands). To minimize variations between assays, frozen serum samples from consecutive visits of the same patient were analyzed on the same plate, using standard curves.

Patients with a baseline concentration ≥ 50 U/mL were considered positive. The distribution of anti-CCP2 titers in the analyzed population is illustrated in Appendix A.

### 2.4. Statistical Analysis

Stata 14.0 for Windows (StataCorp LP, College Station, TX, USA) was used for statistical analysis. Normally distributed quantitative variables were represented as the mean (±standard deviation: SD), while non-normally distributed variables were represented as the median and interquartile range (IQR). Qualitative variables were described using a calculation of the proportions. Variables with a normal distribution were analyzed by t-test, while the Mann–Whitney or Kruskal–Wallis tests were used for variables with a non-normal distribution. To analyze paired samples, the Wilcoxon test was used. The Cuzick’s test, an extension of the Wilcoxon rank-sum test, was used to determine the statistical significance of the distribution trend across ordered groups in variables such as follow-up across visits. A χ^2^ or Fisher’s exact test was used to compare categorical variables. Correlations were analyzed using Spearman’s correlation test. 

After bivariate analysis to identify independent factors that influenced anti-CCP2 levels during the follow-up, we fitted population-averaged models by generalized linear models nested by patient and visit using the xtgee command of Stata. Since anti-CCP2 values did not follow a Gaussian distribution (Appendix A), we used square root transformation of these values in order to achieve a distribution closer to normality. The population-averaged generalized estimating equations were first modeled by adding all variables with a *p*-value < 0.15 in the bivariate analysis. The final models were constructed using quasi-likelihood estimation based on the independence model information criterion [45] and Wald tests, removing all variables with *p* > 0.15.

Subsequently, to analyze which variables had a proportionally higher influence on anti-CCP2 levels, we estimated standardized coefficients by repeating the multivariate analysis with new standardized variables generated with the *egen* command of Stata with the option std, which produces variables with mean 0 and standard deviation 1.

## 3. Results

### 3.1. Study Population 

The study sample comprised 130 patients, whose main sociodemographic characteristics and baseline clinical data are described in Table 1. Briefly, 86% of the patients were women, and the mean age at baseline was 53 years (SD: 15) with a median disease duration of 6 months (IQR: 3.6–9). Baseline disease activity and disability on average were moderate (Table 1). Half of the patients had never smoked and the remaining were active (26%) or former smokers (23%; Table 1). At baseline, a total of 108 patients (83%) fulfilled the 2010 ACR/EULAR criteria for RA classification, while 22 (17%) were classified as undifferentiated arthritis (UA). At the end of the follow-up, 120 (93%) were classified as RA and 10 (7%) as UA.

DMARDs used throughout the follow-up are shown in Table 2. The most frequently used DMARD was methotrexate (MTX) followed by leflunomide and antimalarials. The use of bDMARDs was first recorded at visit 3 (first year of follow-up) when approximately 9% of patients were receiving this kind of therapy. Then, the use of bDMARDs progressively increased up to 21% after 5 years of follow-up. The use of DMARD combination therapy increased along the follow-up until reaching 54% (Table 2).

### 3.2. Baseline Anti-CCP2 Levels and Their Evolution along the Follow-Up

Absolute levels of anti-CCP2 titers showed a sharp decrease from baseline to visit 2 (6 months) and then remained in a plateau until the end of follow-up (Figure 1A). In fact, significant differences compared to baseline were observed at all visits (V2, *p* = 0.0002; V3, *p* = 0.0049; V4, *p* = 0.0001; and V5, *p* = 0.0137). The median (25th; 75th percentile) relative changes from baseline were −47.6% (−12.3; −68) after 6 months (*n* = 94), −43.8% (−12.4; −68) at 1 year (*n* = 93), −49.3% (−14.9; −72) at 2 years (*n* = 97) and −62.3% (−37.9; −71.8) at 5 years (*n* = 48). However, no statistically significant differences were observed in anti-CCP2 values between two to five visits. Figure 1B,C shows how disease activity followed a similar profile of improvement with a sharp decrease from baseline to visit 2 and then a milder decrease of disease activity along the remaining visits.

Then, we analyzed whether the magnitude of decrements in anti-CCP2 levels (Δanti-CCP2) between baseline and 6-month visits correlated with the improvement in disease activity (ΔDAS28, ΔHUPI) in the same period. Figure 2A,B shows that there was an almost significant correlation between these variables, although the correlation coefficient was moderate. In addition, Figure 2C,D shows that the greatest change in anti-CCP2 levels could be observed in those patients with a good response to treatment, although the differences did not reach statistical significance.

Taking all together, these data suggest that although improvement of disease activity can be associated with improvement in anti-CCP2 titers, some other factors may influence the evolution of their titers.

### 3.3. Factors Influencing Anti-CCP2 ACPA Titers

The model in which the level of disease activity at each visit was assessed with HUPI (Table 3) showed a better adjustment than that including DAS28 (Table 4), but the results were quite similar. 

As expected, active smoking was associated with significantly higher levels of anti-CCP2, while in the case of being a former smoker only a non-significant trend was observed (Table 3 and Table 4). Interestingly, the level of disease activity also showed a significant association with anti-CCP2 levels. In fact, visits in which patients showed mild, moderate, or high activity were associated with a progressive and significantly higher level of anti-CCP2 titers, compared with visits where patients were in remission (Table 3). Interestingly, after adjustment by these variables, most treatments were significantly associated with a decrease in anti-CCP2 titers, except for glucocorticoids, antimalarials, and Rituximab (RTX) (Table 3). Nevertheless, the association did not reach statistical significance for RTX, likely due to scanty data restricted to 2 visits in just one patient treated with this antibody (Table 2). 

The results were similar when DAS28 was used to evaluate the activity, but in this model statistical significance was lost for some variables (Table 4). A possible explanation for this difference is that more visits with DAS28 ESR assessment were missed when ESR values were not available, while HUPI can be calculated with either ESR, PCR, or both [43]. 

To explore the impact of combined therapy, we run the multivariate analysis including the variable DMARD treatment with the options: No DMARD (reference), monotherapy, and combination therapy. Glucocorticoids were maintained in the model. Patients on monotherapy showed lower CCP-2 levels than no DMARD (β coefficient −3.748979, 95% CI −5.542862 to −1.955096. *p* = 0.000) and combined therapy showed almost two times lower CCP-2 levels than monotherapy (β coefficient −7.253515 1.128608, 95% CI −9.465546 to −5.041484, *p* = 0.000). Glucocorticid remains non-significant and regarding disease activity levels the β coefficients obtained in this new model (data not shown) were very close to those shown in Table 3. Therefore, we consider the model in Table 3 is more accurate than the last one, with enough information to adjust the situation of DMARD combination and provide individualized information for each DMARD.

Since in both multivariate models the use of bDMARDs was included as a categorical variable whilst csDMARD and glucocorticoids were included as a continuous variable (mg by day or week), the β coefficients are not comparable, giving a false impression that some drugs produce a greater effect on anti-CCP2 levels than others. In order to fix this bias to determine which variables had a more relevant effect on anti-CCP2 levels, the model shown in Table 3 was run using standardized variables as described in the Methods section. The β coefficients obtained and their 95% confidence interval are shown in Figure 3. This approach allowed us to compare the relevance of each variable affecting anti-CCP2 titers.

As shown in Figure 3, being an ever smoker is the variable that exerted the greatest impact on anti-CCP2 levels. The second most important variable, in terms of the magnitude of effect, was disease activity, followed by the different treatments, which did not show significant differences between them.

## 4. Discussion

In recent years, many studies have demonstrated the importance of ACPA in the pathogenesis, prognosis, and clinical outcomes of RA, but the impact of their fluctuations on disease activity and clinical outcomes is still controversial. 

The main contribution of our study is showing that, in patients with seropositive early disease, effective control of disease activity in a clinical practice setting is the main factor accounting for the reduction of anti-CCP2 titers, irrespective of treatment used. Active or ever smoking was the major determinant of high ACPA titers. After adjusting for this habit and disease activity, both cs or bDMARDs independently contributed to an early and significant decline in anti-CCP2 titers that subsequently was maintained throughout five years of follow-up. Therefore, despite a close correlation with disease activity, monitoring ACPA levels over time has no additional benefit over aiming for remission to guide treatment decisions.

Our findings partially agree with previous results in observational studies that examined changes in anti-CCP under csDMARDs, although their timeframe for ACPA measurements was usually shorter [22,26,32]. In accordance with us, Bohler et al. demonstrated that RF and ACPA levels decreased significantly after 6 months of therapy in 143 double positive (RF + anti-CCP) established RA patients, and those reductions were closely linked to an improvement of disease activity [22]. Similar findings were previously reported by Mikuls et al. in 66 early RA patients that achieved substantial reductions in disease activity in response to MTX, sulfasalazine (SSZ), or hydroxychloroquine (HCQ); however, in adjusting models, only a shorter disease duration (≤12 months) was significantly associated with a decline in anti-CCP titers at 6 months, while no association with specific drug or treatment response was observed [26]. By contrast, in the early seropositive RA cohort studied by Ally et al., significant drops in anti-CCP levels detected 6 months after therapy with csDMARDs (mostly MTX) did not correlate with changes in disease activity [32]. 

Only one early longitudinal study explored the fluctuations of anti-CCP titers for a five-year follow-up [34], as in the work described herein. A significant drop in anti-CCP levels was reported during the first year, which correlated with the extent of treatment with SSZ but not with other csDMARDs or clinical activity [34]. In contrast with the stabilization of anti-CCP titers after the initial decline in ours and other studies [36], significant rises were observed after 1, 2, and 5 years. We could only hypothesize that a worse clinical disease course in seropositive compared to seronegative patients could explain these differences, as no data on correlations between disease activity and anti-CCP levels were shown beyond one year.

Taking into account all previous data, the inconsistent association of ACPA decline and changes in disease activity or clinical outcomes could be explained by the heterogeneity in study populations, disease duration, ACPA measurement assays and study periods, csDMARDs usage, and therapeutic strategies. 

An absence of this consistent correlation has also been described in studies including different bDMARDs [27,33,36]. It is important to note that most evidence comes from observational studies, in established RA populations or focused on baseline ACPA seropositive status as a predictor of response [27]. The association between decreases in ACPA levels and reduction in disease activity has been distinctively reported for ABA compared to adalimumab (ADA) in established RA [46]. In seropositive early RA, posthoc analysis of the AVERT [47] and AGREE [48] trials revealed a greater decrease of anti-CCP levels or seroconversion induced by ABA + MTX versus MTX monotherapy, followed by better clinical [49] and radiological outcomes than patients on MTX alone, or patients on either arm which remained ACPA positive [25]. Those effects are consistent with the ABA mechanism of action, through the inhibition of costimulatory signals that secondarily affect B lymphocytes and plasma cells responsible for antibody production.

Interestingly, data from two other early RA trials with step-up therapy guided by the 2T2 strategy supported that intensity of treatment, irrespective of specific drug usage, is associated with the decline in anti-CCP titters, but differs in its association with clinical response. [33,36]

Our results are closer to the findings reported by Kastbom et al., in the SWEFOT study [36]. In this early RA trial, inadequate responders to MTX monotherapy at 3 months were randomized to add-on therapy with SSZ plus HCQ, or infliximab (IFX). A pronounced decline in median levels of all tested ACPAs was detected during the first 3 months of MTX therapy and then for subsequent 3–12 months following response to add-on therapy, with no significant association to treatment regimen [36]. Also close to our data, the magnitude of anti-CCP decline during add-on therapy was significantly higher in responders, though the trend we observed for EULAR response at 6 months did not achieve statistical significance. 

Although our study analyzed data from real-life practice, the use of DMARDs is quite similar to the approach in the SWEFOT trial. At 6 months, most patients were under MTX monotherapy, and then, following a non-protocolized 2T2 approach, the use of combined therapy either with cs or bDMARDs gradually increased (Table 2), accounting for substantial reductions in disease activity. 

The design in the second trial differed as treatment could be either tapered or escalated [33]. Following initial prednisone and MTX, treatment was changed every 4 months aiming for DAS remission (add-on to triple therapy or ADA) or drug-free remission (de-escalation). In contrast with ours and SWEFOT findings, de Moel et al. observed that fluctuations of anti-CCP titers did not correlate with disease activity over 1 year but were in close relationship with escalation (decline) and de-escalation treatment (rises), and therefore with the intensity of treatment. Those findings do not invalidate ours, as they showed that irrespective of disease activity or the treatment used, ACPA levels are modulated by commonly used DMARDs.

Concerning that, the longitudinal multivariate model nested by patient and visit allowed us to estimate the effect of different DMARDs and glucocorticoids. Adjusting by smoking habit and level of disease activity, we demonstrated that almost all cs and bDMARDs independently influence the reduction of anti-CCP2 titers in patients with EA. MTX showed the highest effect and antimalarials and glucocorticoids did not exert any significant effect. Regarding antimalarials, besides a reduced use in our population, a less relative effectivity of this drug compared to other DMARDs is well-known on RA and can likely explain a modest impact in reducing disease activity. A greater non-significant effect was observed for glucocorticoids, which were prescribed in less than 30% of patients at baseline, mostly as a bridging therapy, and doses rarely exceeded 7.5 mg/day dose. Therefore, low doses and use in a restricted population in the early period when we observed the most pronounced drop in anti-CCP2 levels could be behind our results. Just one patient was on RTX, which likely precluded to reach statistical significance. Also, those data reinforce the findings in other early RA cohorts [32,34], with significant reductions in ACPA levels irrespective of disease activity.

Our adjusted model also showed that active smoking was the greatest determinant of high ACPA titers, in line with previous descriptions in established RA [50], although this association was not demonstrated in two studies on early RA [32,33].

One potential limitation of our study is the small numbers of patients under bDMARDs, specially RTX, which have likely precluded to demonstrate a statistically significant effect on the anti-CCP decline, although subsequent adjustment with β standardized coefficients revealed a more realistic magnitude of effect. Secondly, some patients do not have serum samples available from all follow-up visits; nevertheless, these patients are not a substantial proportion of the population, which remains homogeneous across the time-points, except for the last visit when just half of the population has reached this follow-up time. Therefore, we think it is unlikely this issue has affected the main results.

Conversely, some strengths of our study can be highlighted. The longitudinal design of our cohort provides fluctuations of ACPA titers for five years of follow-up in an EA population that overall had not received DMARDs at recruitment. The protocolized data collection precludes substantial missing data and accounts for all RA medications received by the patients during the study period. Lastly, in contrast with previous studies, we have provided data on the impact of each DMARD on the modulation of anti-CCP levels in a real word setting.

## 5. Conclusions

Our results show that in patients with seropositive EA treated in a real practice setting, an early and significant drop in anti-CCP2 titers can be detected associated with the decline in disease activity, irrespective of the treatment used. TNF antagonists and cs DMARDs, except antimalarials, can also independently contribute to a sustained decline in titers that is maintained throughout five years of follow-up.

## Figures and Tables

**Figure 1 diagnostics-12-01773-f001:**
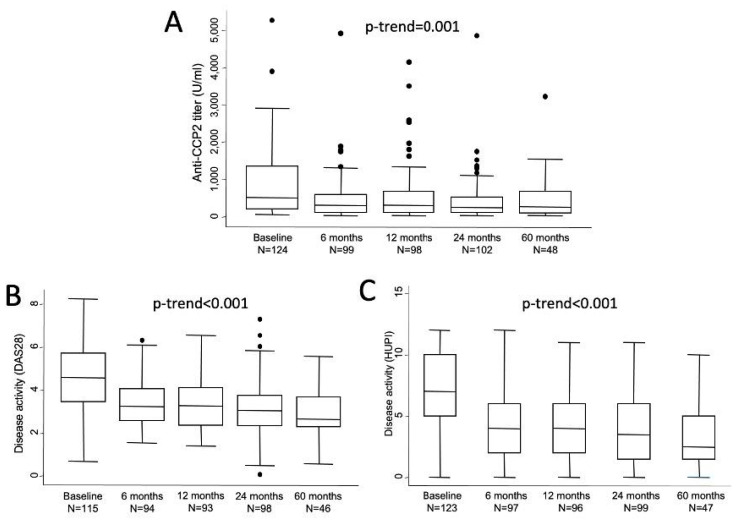
Evolution of anti-CCP2 titers (**A**) and disease activity assessed either with DAS 28 (**B**) or with HUPI (**C**). Data are shown as interquartile ranges (p75 upper edge of box, p25 lower edge, p50 midline) as well as the p95 (line above box) and p5 (line below). Dots represent outliers. Statistical significance was determined with the Mann-Whitney test.

**Figure 2 diagnostics-12-01773-f002:**
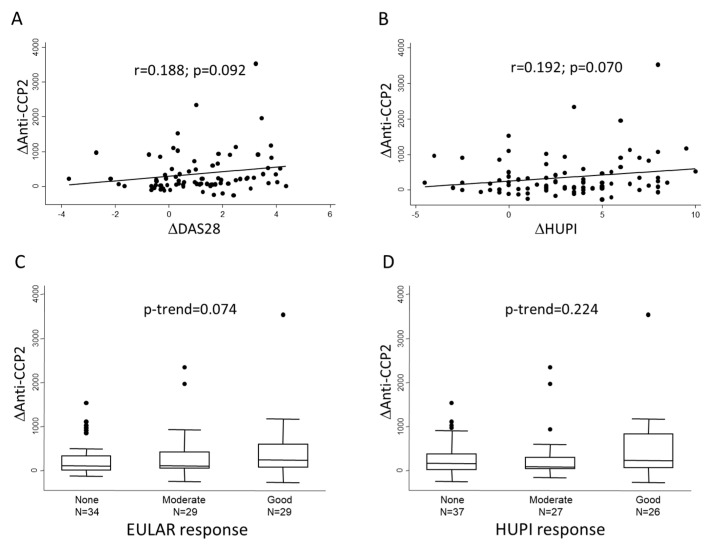
Association between change in anti-CCP2 levels and therapeutic response at six months of follow-up. Panels (**A**,**B**) show the lineal correlation between change in anti-CCP2 levels (Δanti CCP-2) and change in disease activity assessed either with DAS28 or HUPI scores, respectively. Panels (**C**,**D**) show Δanti-CCP2 distribution according to categories of response; in this panels dots represent outliers. Statistical significance was assessed through Spearman’s correlation test in panels (**A**,**B**) and with Cuzick’s test in panels (**C**,**D**).

**Figure 3 diagnostics-12-01773-f003:**
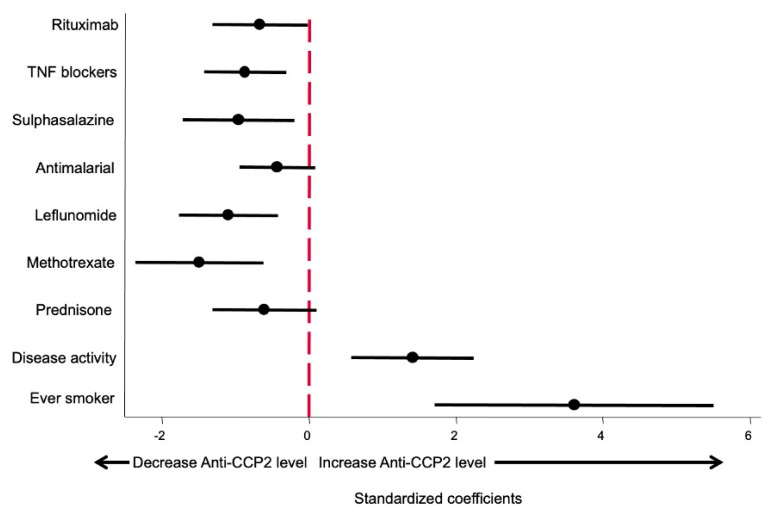
Effect size of different variables on anti-CCP2 ACPA titers. Data are displayed as the β coefficients, and their 95% confidence interval was obtained by running the model shown in Table 3 but using standardized variables (see Methods section).

**Table 1 diagnostics-12-01773-t001:** Characteristics of the population at recruitment (Visit 1).

Sex *n* (%)	Male	18 (13.85%)
	Female	112 (86.15%)
Diagnosis at baseline (ACR/EULAR Criteria 2010) *n* (%)	Rheumatoid Arthritis	108 (83.08%)
	Undifferentiated arthritis	22 (16.92%)
Age at onset *n* (%)	<45 years	42 (32.31%)
	45–65 years	58 (44.62%)
	>65 years	30 (23.08%)
	Mean (SD)	52.64 (15.12)
Race, ethnicity *n* (%)	Caucasian	108 (83.08%)
	Latin American	22 (16.92%)
Smoking *n* (%)	Never-smoker	65 (50.78%)
	Former smoker	30 (23.44%)
	Active smoker	33 (25.78%)
Time of evolution to recruitment (weeks)	Median (IQR)	6 (3.6–8.97)
Baseline anti-CCP2 titers	Median (IRQ)	460.3 (160–1280)
Baseline DAS28	Median (IQR)	4.53 (3.47–5.63)
Baseline HUPI	Median (IQR)	7 (5–10)
Baseline HAQ	Median (IQR)	1 (0.5–1.63)
Baseline CRP	Median (IQR)	0.7 (0.2–1.5)

ACR: American College of Rheumatology; CCP: Cyclic citrullinated peptide; CRP: C-reactive protein; DAS28: Disease activity score 28 joints count; HAQ: Health Assessment Questionnaire; HUPI: Hospital Universitario la Princesa index; IQR: Interquartile range; SD: standard deviation.

**Table 2 diagnostics-12-01773-t002:** DMARD treatment prescribed at each visit.

Drug	Visit 1 (*n* = 124)	Visit 2 (*n* = 98)	Visit 3 (*n* = 97)	Visit 4 (*n* = 102)	Visit 5 (*n* = 48)
Glucocorticoids	27.63%	54.07%	45.36%	25.49%	22.92%
Methotrexate	19.35%	83.67%	75.26%	70.59%	68.75%
Leflunomide	0.81%	11.22%	22.68%	23.53%	31.25%
Anti-malarials	4.84%	17.35%	20.62%	17.65%	18.75%
Sulphasalazine	0.81%	5.10%	4.12%	6.86%	10.42%
Gold salts	0.81%	1.02%	2.06%	1.96%	0%
Anti-TNF	0%	1.02%	5.15%	5.88%	18.75%
Tocilizumab	0%	0%	0%	1.96%	0%
Rituximab	0%	0%	0%	0.98%	2.08%
Combination therapy	3.08%	23.21%	29.2%	37.4%	53.97%

DMARD: Disease-modifying anti-rheumatic drugs; TNF: Tumor necrosis factor.

**Table 3 diagnostics-12-01773-t003:** Multivariate analysis showing variables associated with the evolution of anti-CCP2 titers. Adjusted model with disease activity assessed by HUPI index.

Independent Variable		β Coefficient	95%CI	*p* Value
Smoking	Never	Reference category		
	Former smoker	4.09	−0.67–8.85	0.092
	Smoker	9.26	4.59–13.92	<0.001
Disease activity (HUPI)	Remission	Reference category		
	Mild	2.30	0.48–4.11	0.013
	Moderate	2.35	0.21–4.48	0.031
	High	3.76	1.18–6.34	0.004
Glucocorticoids (mg/day)		−0.10	−0.24–0.41	0.162
Methotrexate Dosage (mg/week)		−0.177	−0.28–−0.075	0.001
Leflunomide Dosage (mg/day)		−0.25	−0.38–−0.11	0.001
Anti-Malarials Dosage (mg/day)		−0.006	−0.12–0.001	0.053
Sulphasalazine Dosage (mg/day)		−0.003	−0.005–−0.001	0.010
Anti-TNF (YES/NO)		−4.97	−8,48–−1.48	0.005
Rituximab (YES/NO)		−9.75	−20.31–0.81	0.070

CI: confidence interval. HUPI: Hospital Universitario la Princesa index. TNF: Tumor necrosis factor.

**Table 4 diagnostics-12-01773-t004:** Multivariate analysis showing variables associated with the evolution of anti-CCP2 titers. Adjusted model with disease activity assessed by DAS28.

Independent Variable		β Coefficient	95%CI	*p* Value
Smoking	Never	Reference category		
	Former smoker	4.11	−0.73–8.95	0.096
	Smoker	8.98	4.21–13.75	0.000
Disease activity (DAS28)	Remission	Reference category		
	Mild	0.69	−1.56–2.94	0.547
	Moderate	2.76	0.85–4.66	0.005
	High	3.70	1.14–6.25	0,005
Corticosteroids (mg/day)		−0.13	−0.27–0.01	0.079
Methotrexate Dosage (mg/week)		−0.17	−0.28–−0.07	0.001
Leflunomide Dosage (mg/day)		−0.22	−0.36–−0.08	0.002
Anti-Malarials Dosage (mg/day)		−0.005	−0.11–0.0008	0.087
Sulphasalazine Dosage (mg/day)		−0.003	−0.005–−0.0005	0.014
Anti-TNF (YES/NO)		−5.20	−8.57–−1.84	0.002
Rituximab (YES/NO)		−10.37	−20.68–−0.06	0.049

CI: confidence interval; DAS28: Disease activity score 28 joints count; TNF: Tumor necrosis factor.

## Data Availability

The data presented in this study are available on request from the corresponding author.

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
