# Peer review of "Anti-Citrullinated Protein Antibody Titers Are Independently Modulated by Both Disease Activity and Conventional or Biologic Anti-Rheumatic Drugs"

_diagnostics, 2022, doi:10.3390/diagnostics12071773_

Round 1
Reviewer 1 Report
Rheumatoid arthritis is an extremely serious disease that affects more and more people. Despite the two most commonly used disease markers, such as RF and aCCP, correct diagnosis of the disease is still a serious problem. Especially in the early stages of the disease. That is why, in my opinion, the research conducted by the authors of the article is so important. However, to make it easier for the reader to read, the title of tables 3 and 4 should be changed. The name of the model should be in the title of the table and not under it, because in this form it is misleading. I believe that the lack of influence of treatment with glucocorticoids and antimalarial drugs on the decrease in anti-CCP2 titers deserves a wider comment.
Author Response
Response: Thank you for your kind appreciation of our work. Following your useful suggestion, we have changed the title in Tables 3 and 4 as follows “Table 3. Multivariate analysis showing variables associated with evolution of anti-CCP2 titers. Adjusted model with disease activity assessed by HUPI index” “Table 4. Multivariate analysis showing variables associated with evolution of anti-CCP2 titers. Adjusted model with disease activity assessed by DAS28”
Regarding the lack of influence of treatment with antimalarials on anti CCP-2 titters, we think it can be likely explained by a well-known less relative effectivity of this DMARD compared to other DMARDs on RA (Felson DT et al. Arthritis Rheum 1992 35(10): 1117) and therefore, less impact in reducing disease activity. Additionally, as shown in Table 2, antimalarials was prescribed in less than 5% of patients at baseline and reached only 17% of patients during the first six months, the period in which we observed the most pronounced drop in anti-CCP titters.
As we have previously reported (Ibañez et al. Arthritis Res Ther 12, R50, 2010), in our cohort, oral GC therapy was mostly prescribed as a bridging therapy and doses rarely exceeded 7,5mg/day dose. In the present work, GC was indicated in less than 30% of patients at baseline, rising to half of patients at 6 months, and then gradually decreases throughout the study visits. We have introduced a sentence in the discussion in line 442-448 to clarify these issues: “Regarding antimalarials, besides a reduced use in our population, a less relative effectivity of this drug compared to other DMARDs is well-known on RA and can likely explain a modest impact in reducing disease activity. A greater non-significant effect was observed for glucocorticoids, that were prescribed in less than 30% of patients at baseline, mostly as a bridging therapy and doses rarely exceeded 7,5mg/day dose. Therefore, low doses and use in a restricted population in the early period when we observed the most pronounced drop in anti-CCP2 levels, could be behind our results.”
Reviewer 2 Report
This is an interesting study by Uriarte Ecenarro et al that recruit 130 ACPA-positive patients from the PEARL study along with a 5-year follow-up. Various DMARDs treatments seem to reduce ACPA titers and disease activity in their cohort, however, there was an almost significant correlation between anti-CCP2 titers and DAS28 or HUPI. They further identified ever smoker as the most important variable that increased anti-CCP levels. Early diagnostics and treatments for RA patients are very important, especially by evaluating ACPA titers in these patients.
Minor revisions:
1. In the conclusion, the authors said that “Both cs and bDMARDs can also independently contribute to a sustained decline in titers that are maintained throughout five years of follow-up”. However, in table 3, glucocorticoids, antimalarials, and Rituximab (RTX) were not significantly associated with a decrease in anti-CCP2 titers. The sentence should be modified.
2. As above, the association did not reach a statistical significance for RTX is very surprising, although they ascribed to a low number of visits in which the patients had been treated with this antibody. Could they provide pieces of literature to explain the phenomenon?
3. Have the authors ever checked RF+anti-CCP in their cohort? As they have mentioned in ref.22, the combined factors were more closely linked to disease activity.
Author Response
Minor revision
1) In the conclusion, the authors said that “Both cs and bDMARDs can also independently contribute to a sustained decline in titers that are maintained throughout five years of follow-up”. However, in table 3, glucocorticoids, antimalarials, and Rituximab (RTX) were not significantly associated with a decrease in anti-CCP2 titers. The sentence should be modified.
Response: Our intention was to conclude that DMARDs, irrespective of being conventional synthetic or biologic class, can reduced ACPA titers, but we agree with the appreciation of the reviewer and have changed the text accordingly “TNF antagonist and cs DMARDs, except antimalarials, can also independently contribute to a sustained decline in titers that are maintained throughout five years of follow-up”
2. As above, the association did not reach a statistical significance for RTX is very surprising, although they ascribed to a low number of visits in which the patients had been treated with this antibody. Could they provide pieces of literature to explain the phenomenon?
Response: Just 1 patient in our cohort in two late visits 4 (2 years) and 5 (five years) was under RTX therapy and we think this is the reason that prevent demonstrating a statistically significant effect.
3. Have the authors ever checked RF+anti-CCP in their cohort? As they have mentioned in ref.22, the combined factors were more closely linked to disease activity.
Response: Thanks for raising this question. We have run the multivariate analysis restricted to the double FR+/ACPA+ population and we could not find any significant differences with the results in the whole population neither in the association with disease activity nor in the effect of treatments.
Reviewer 3 Report
The authors evaluated the effects of disease activity and treatments on CCP-2 levels.
1) Please indicate the reference of the PEARL study in the article
2) Did the authors evaluate patients' drug compliance using standard methods?
3) Authors may also add the effect of combination therapy on CCP-2 levels
4) Also, steroid dose can be an independent factor for changes in CCP-2 levels. The authors can categorize patients according to steroid dose.
5) Why do the authors not include the effect of non-TNF biologics on CCP-2 other than rituximab?
6) How can we use this data in our daily practice? Do the results suggest that we can use CCP-2 as a follow-up marker for disease activity or patient adherence?
Author Response
1) Please indicate the reference of the PEARL study in the article
Response: The reference is number 39, cited in the text in lines 102 and 103
2) Did the authors evaluate patients' drug compliance using standard methods?
Response: This is a study conducted in a clinical practice setting, thus, unfortunately we have not evaluated patients´drug compliance in a standardised manner
3) Authors may also add the effect of combination therapy on CCP-2 levels
Response: Thank you for raising this interesting issue. We have run the multivariate analysis including the variable DMARD treatment with the options: No DMARD (reference), Monotherapy and Combination Therapy. This new model was run excluding the individual DMARDs, since there is a huge collinearity between them and the new variable. Glucocorticosteroids were maintained in the model. As you can see, below patients in monotherapy showed lower CCP-2 levels than no DMARD, and Combined Therapy showed almost two times lower CCP-2 levels than Monotherapy.
Glucocorticosteroid remain non-significant and regarding disease activity levels the beta coefficients obtained in this new model were close to those showed at Table 3.
We think the model showed in Table 3 is more accurate than this one, since considering the generalized linear models nested by patient and visit, there is enough information to adjust the situation of DMARD combination and it provides individualized information for each DMARD.
Nonetheless, we have added a new paragraph in lines 302-312 of the revised manuscript to describe those results: "To explore the contribution of combined therapy, we run the multivariate analysis including the variable DMARD treatment with the options: No DMARD (reference), monotherapy and combination therapy. Glucocorticoids were maintained in the model. Patients on monotherapy showed lower CCP-2 levels than no DMARD (β coefficient -3.748979, 95% CI -5.542862 to -1.955096. p=0.000) and combined therapy showed almost two times lower CCP-2 levels than monotherapy (β coefficient -7.253515 1.128608, 95%CI -9.465546 to -5.041484, p=0.000). Glucocorticoids remain non-significant and regarding disease activity levels the β coefficients obtained in this new model (data not shown) were very close to those showed at Table 3. Therefore, we consider the model in Table 3 is more accurate than this last one, with enough information to adjust the situation of DMARD combination and providing individualized information for each DMARD".
4) Also, steroid dose can be an independent factor for changes in CCP-2 levels. The authors can categorize patients according to steroid dose.
Response: In our multivariate models, the effect of glucocorticoids has not been analyzed as a dicotomic variable (YES/NO). The beta coefficient shows the effect per mg of prednisone. Additionally, in our cohort, doses rarely exceed 5-7,5mg/d of prednisone or equivalent. Therefore, we think a stratified dose analysis would not add additional relevant information to explore the effect on ACPA titers
5) Why do the authors not include the effect of non-TNF biologics on CCP-2 other than rituximab?
Response: As you can see in Table 2, just 2 patients in one visit were under tocilizumab therapy. In the generalized linear models nested by patient and visit, this drug was excluded from the model as no significant effect could be found in the bivariate analysis.
6) How can we use this data in our daily practice? Do the results suggest that we can use CCP-2 as a follow-up marker for disease activity or patient adherence?
Response: As we have state in the discussion, our main finding is that effective control of disease activity in a clinical practice setting is the main factor accounting for the reduction of anti-CCP2 titers, irrespective of treatment used. The most pronounced drop in these levels occurs early (fig 1A) in close correlation with the decrease in disease activity and remains throughout the follow-up if disease control is sustained. Therefore, to guide treatment decisions, monitoring ACPA levels over time has no additional benefit over aiming for remission.
The variations of ACPA titers in relation with patients´ adherence, rises an interesting question that deserves additional research. Unfortunately, we do not monitor patient adherence with standardized tools in the real clinical practice. Monitoring anti-CCP2 titers to assess compliance with medications could be useful is these medications have shown effective in an individual patient and suddenly the disease control is loss. However, the design of studies to answer this query should take into account multiple confounding factors.